# Obesity and Thyroid Axis

**DOI:** 10.3390/ijerph18189434

**Published:** 2021-09-07

**Authors:** Krzysztof Walczak, Lucyna Sieminska

**Affiliations:** 1Department of Thoracic Surgery, Faculty of Medical Sciences in Zabrze, Medical University of Silesia in Katowice, 41-800 Zabrze, Poland; krzysiekmed@gmail.com; 2Department of Pathophysiology and Endocrinology, Faculty of Medical Sciences in Zabrze, Medical University of Silesia in Katowice, 41-800 Zabrze, Poland

**Keywords:** adipose tissue, obesity, thyroid, TSH, adipokines

## Abstract

Development of obesity is primarily the result of imbalance between energy intake and energy expenditure. Thyroid hormones influence energy expenditure by regulating cellular respiration and thermogenesis and by determining resting metabolic rate. Triiodothyronine influences lipid turnover in adipocytes and impacts appetite regulation through the central nervous system, mainly the hypothalamus. Thyroid-stimulating hormone may also influence thermogenesis, suppress appetite and regulate lipid storage through lipolysis and lipogenesis control. Subclinical hypothyroidism may induce changes in basal metabolic rate with subsequent increase in BMI, but obesity can also affect thyroid function via several mechanisms such as lipotoxicity and changes in adipokines and inflammatory cytokine secretion. The present study investigated the complex and mutual relationships between the thyroid axis and adiposity.

## 1. Introduction

Thyroid hormonal tests are routinely ordered when seeking obesity origins [1]. Often higher levels of serum thyrotropin (thyroid-stimulating hormone, TSH) within the reference range or slightly elevated levels are found in the obese state [2,3,4]. In the past decades, a number of reports tried to explain whether thyroid dysfunction is the cause or rather the consequence of excess adipose tissue; the answer, however, remains unclear.

The relations between the hypothalamic–pituitary–thyroid axis (HPT) and obesity are complex and include different interactions. Thyroid hormones (THs) and TSH independently regulate the mass and function of adipose tissue; however, adipose tissue, through the production of adipokines (adipocytokines), also affects the activity of the HPT system. The objectives of this review were (1) the presentation of fundamental mechanisms linking the HPT axis with energy balance, (2) characterization of factors modifying primary relationships and (3) presentation of complicated relationships in a manner helping a physician evaluating the thyroid axis of obese patients to conclude if optimizing the thyroid axis is desirable.

THs regulate the function of many tissues and organs—heart, liver, brain, skeletal muscles, pancreas and adipose tissue. They control the energy balance, appetite, basal metabolic rate (BMR), thermogenesis, free fatty acid (FFA) oxidation, glucose and lipid metabolism [5,6,7]. The actions of THs in tissues are determined by transmembrane transport, intracellular deiodinase (DIO) activity and expression of TH receptors (TRs). Free THs can bind to membrane-bound receptors or can be transported from the extracellular space into the cytosol through specific transporters: monocarboxylate transporter 8 (MCT8) and organic anion-transportingpolypeptide 1c1 (OATP1c1). Adrenergic receptors and signaling also play an important role. The direct evidence for the influence of THs on adipose tissue is the expression of TRs in human adipocytes. Triiodothyronine (T3) acts through TRs: TRα-1, TRα-2, TRβ-1 and TRβ-2. The receptors are encoded by two genes: TRα and TRβ [8,9]. T3 is considered to be the active form of the hormone, and T4 is mainly its precursor. As a specific endogenous ligand, T3 modulates the activity of nuclear and mitochondrial TRs. It was demonstrated that THs exert both direct action on transcription (canonical regulation of gene transcription, called genomic) and non-genomic effects, mediated by cell membrane and mitochondrial binding sites (non-genomic activity). Recently, it was shown that 3,5-diiodo-L-thyronine (3,5-T2) derived from T3 by deiodination has a positive effect on energy metabolism and lipid metabolism. 3,5-T2 increases the rate of consumption by mitochondria, the BMR and prevents high-fat-diet-induced obesity [10].

The main site of thyrotropin action is the thyroid gland, and TSH works by binding to its receptor G-protein-coupled TSH receptor (TSHR). In addition to thyroid tissues, the presence of TSHR was reported in some non-thyroidal tissues including liver, ovary, adipocytes, immune cells, ocular muscles and erythrocytes; however, its role in extra-thyroid tissues remains unclear. It was assumed that TSH/TSHR signaling plays a role in the maintenance of the proper function of adipose tissue and TSH could affect thermogenesis, adipogenesis and lipolysis/lipogenesis balance. Several in vitro experiments demonstrated the presence of TSHR in mouse and rat preadipocytes and that TSHR expression increased with preadipocyte differentiation [11,12]. Functional TSHRs were expressed in rat brown adipose tissue (BAT), and TSH increased the expression of DIO2 and uncoupling protein type 1 (UCP-1) [13]. Studies have revealed the presence of TSHRs in human white and brown adipocytes [14,15,16,17,18,19,20,21].

Bell and coworkers found the expression of TSHR mRNA in subcutaneous abdominal tissue [16]. They also described the presence of functional TSHR protein in human preadipocytes isolated from abdominal subcutaneous and omental tissue. TSH treatment of preadipocytes induced activation of the p70 S6 kinase, a downstream target of TSHR, independently of cAMP-mediated TSH signaling [16]. TSHR transcripts in human adult abdominal adipose tissue were also detected by Crisp et al. [14]. In addition, TSHR protein expression in the human subcutaneous adipose tissue was determined in 120 patients with different BMIs by Lu et al. [15]. The TSHR expression tends to increase with increasing BMI. In the study by Janson et al. [20], TSHR RNA expression in adipose tissue and isolated adipocytes obtained from infants, children and adults was demonstrated. The expression seemed to be age-dependent as it was higher in infants than adult adipose tissue. In the study by Nannipieri et al. [21], TSHR expression was shown to decrease in obesity.

The generation of new adipocytes by differentiation of fibroblast-like preadipocytes into mature adipocytes, under the influence of various transcription factors, is called adipogenesis. TSH-TSHR signaling plays a regulatory role in this process. TSHR mRNA levels in mature adipocytes were 100-fold greater than in preadipocytes in the in vitro experiment [22]. It is important to realize that, in certain situations, adipogenesis slows down. Various factors reduce or enhance expression of TSHR. Some mechanisms are understood, and in the following sections, we present complex functions regulating TSH-TSHR signaling.

TSHR activation occurs not only at elevated TSH but also in Graves’ disease when thyroid-stimulating autoantibodies (TSAb) activate TSHR on orbital fibroblasts and T cells. In Graves’ disease, TSHR enhances adipogenesis not only in orbital adipose tissue but also in white adipose tissue [18]. Some researchers speculate that TSHR activation by TSAb affects body fat composition in Graves’ disease patients; however, the final effect depends on thyroid status. In hyperthyroid patients, excess of THs and stimulation of TSHR by TSAb promotes adipogenesis and brown adipose tissue formation, and therefore energy can be dissipated as heat and patients lose weight. Persisted elevated TSAb in patients with Graves’ disease and posttreatment euthyroidism favors adipogenesis in white adipose tissue and leads to weight gain.

TSH consists of two subunits: a common α subunit and a unique β subunit. Recently, TSH β (TSHB) gene mRNA and protein expression in both visceral and subcutaneous human adipose tissue were described; however, TSH alpha subunit mRNA and protein were not demonstrated [22,23]. TSHB mRNA was positively correlated with the expression of mitochondrial function and fatty-acid-mobilization-related genes which suggests involvement in energy expenditure [23]. Adipose-tissue TSHB protein level was positively correlated with mitochondrial respiratory capacity only in the visceral depot. Interestingly, no association was found between TSH protein and TSHB gene expression [23]. Moreover, adipose-tissue TSHB mRNA was significantly correlated with serum total and LDL-cholesterol levels in three independent groups of patients (cohort 1 = 38 patients with mean BMI 45.1 ± 6.4 kg/m^2^, cohort 2 = 73 subjects with BMI between 19.5–85.2 kg/m^2^, cohort 3 = 20 participants with mean BMI 42.8 ± 5.5 kg/m^2^). The results provide evidence that TSHB is implicated with cholesterol metabolism in human adipose tissue. Both visceral and subcutaneous adipose tissue TSHB correlated positively with mRNA of 3-hydroxy-3-methylglutaryl-CoA reductase (HMGCR), the key enzyme in cholesterol biosynthesis. No associations were found between serum total and LDL cholesterol and TSHR expression in adipose tissue [22]. Administration of recombinant human TSH resulted in increased HMGCR mRNA levels in fully differentiated adipocytes but not in preadipocytes [22]. Treatment of human mature adipocytes with recombinant TSH enhanced mitochondrial respiratory capacity and ATP production and led to increased adipogenesis-related genes, but no effects were observed in human preadipocytes with low expression of TSHR [23]. As an inverse correlation between thyroxine and TSHB mRNA in adipose tissue was found, Moreno-Navarrete et al. suggested that local TSH production may be related to THs [22]. Moreover, researchers demonstrated that the main source of TSHB was stromal vascular cell fraction. All the observations make the detailed role of TSHB in adipose tissue not fully understood, but it seems likely that TSHB acts as a paracrine factor that is implicated in lipid metabolism and energy homeostasis. In order to explore specific mechanisms linking TSHB and adipocyte function, further studies are needed.

Reports suggest that thyrotropin-releasing hormone (TRH) and TSH play a role in the regulation of appetite. Central administration of TRH and TSH in experimental animals caused a reduction in food intake, and similar effects were seen following peripheral administration of TRH (Table 1) [24].

## 2. Different Depots of Adipose Tissue

There are two main types of fat tissue—white adipose tissue (WAT) and BAT—which serve opposite functions in energy balance regulation. WAT includes subcutaneous and visceral deposits, and the latter can grow as mesenteric, omental, retroperitoneal, epicardial and perivascular fat. WAT is composed mainly of white adipocytes and preadipocytes, but it also contains several other cells such as macrophages, lymphocytes, fibroblasts, stromal cells and extracellular matrix. WAT plays a role as an energy reservoir in the form of triglycerides and as an important endocrine organ that produces adipocytokines. White adipocytes specialize in lipid storage in response to energy status and can mobilize energy as FFAs if necessary. WAT has the ability to change volume and can expand or contract. It grows by two mechanisms: hyperplasia (increased number of fat cells) and hypertrophy (increased adipocyte volume). The size of adipocytes depends on adipocyte triglyceride storage which is the final result of the balance between lipogenesis, lipolysis and fatty acid oxidation. These processes are tightly regulated by insulin and catecholamines; however, TSH and THs also play a role (Figure 1). This phenomenon may evolve with age, and it may be affected by other factors.

Under the influence of transcription factors, such as peroxisome proliferator-activated receptor γ (PPARγ) and CCAAT/enhancer-binding proteins (C/EBPs), generation of new adipocytes through differentiation of progenitor cells occurs (adipogenesis). This process is also under TSH/THs control [5] (Figure 2). Increased adipocyte size correlates with impaired adipogenesis [5].

It has long been thought that BAT occurs only in newborns and then disappears. However, recently, it was shown that BAT is present also in adults, and its high activity is associated with a favorable metabolic profile [5,6,7]. The basic function of BAT is heat production in a process called thermogenesis.

## 3. Thermogenesis

The heat generated during vital functions sufficient to maintain body temperature is called obligatory thermogenesis. THs may increase obligatory thermogenesis by increasing the BMR. THs also control adaptive (facultative) thermogenesis mediated by BAT. Isolated signaling of THs in dissipating energy is insufficient, and to provide an adequate effect synergy of both THs and adrenergic signaling is necessary. THs regulate non-shivering thermogenesis by targeting BAT, WAT, skeletal muscle and skin blood flow.

Unlike white adipocytes, which store lipids as a single large droplet, brown adipocytes contain many smaller droplets. They are characterized by numerous mitochondria that express the specific UCP1 and are richly innervated by sympathetic nerves. The UCP1 is responsible for energy dissipation as heat, and its activation is controlled by adrenergic activity and by the concentration of T3 in adipocytes [5,6]. Under sympathetic stimulation, β3-adrenergic receptors are activated, and FFA-releasing lipolysis begins. Recently it was reported that thermogenesis in human BAT is mediated by the stimulation of β1-AR and β2-AR [25]. Cold stress via stimulation of the sympathetic nervous system and by increasing norepinephrine (NE) in BAT acutely stimulates BAT-specific T3 production and mitochondrial heat generation. Adrenergic stimulation increases the activity of DIO2 and the intracellular conversion of thyroxine (T4) to T3 that directly stimulates UCP1 expression via TH-response elements (TREs) located in the 5′-flanking region of UCP1 [26]. This phenomenon occurs in BAT within hours after acute cold stress; however, during persistent cold exposure, the remodeling of WAT takes place, during which mature adipocytes begin to acquire the thermogenic capability.

T3 acts synergistically with NE released from sympathetic nerve endings to stimulate heat production. Increased local production of T3 from T4 in BAT via sympathetic activation and DIO2 expression has been well documented. Both DIO2 and UCP-1 are the indexes of thermogenic activation of BAT [5,6].

Not only THs but also TRs are essential for adequate thermogenesis in BAT. Brown adipocytes possess both TRα1 and TRβ nuclear receptors. While TRα1 is necessary for the proper adrenergic stimulation of brown adipocytes, TRβ isoform is responsible for T3-regulated UCP1 activation that was confirmed in TRβ gene knockout mice [27,28]. Locally generated T3 by conversion of T4 by DIO2 is required for saturation of TRs [29].

The combined synergistic actions of both T3 and NE can efficiently induce UCP1 expression; therefore, in different pathological conditions with deficits or excess of THs, this mechanism could be disturbed. Whereas cold-induced adaptive thermogenesis plays a role when thyroid function is normal, in hypothyroid rats, UCP1 expression is lower and could not be upregulated during cold exposure [30]. In humans, it is well known that BAT thermogenesis is reduced in hypothyroidism and patients often report cold intolerance. It is worth mentioning that mechanisms partially compensating for the reduction of thermogenesis exist. Experimental models revealed that animals with mild hypothyroidism display elevated BAT adrenergic sensitivity [30,31]. Increased sympathetic tone in a compensatory manner can activate BAT and increase heat production. Thus, sympathetic activity seems to partially compensate for diminished THs signaling. However, when hypothyroidism reaches a significant level, the above mechanism collapses, which can be explained by the central regulation of this phenomenon [28].

Weiner et al. described another compensatory response [32]. The decreased metabolic activity of BAT in hypothyroidism was associated with a strong increase of thermogenesis in BAT and with significant upregulation of the adrenergic system. The authors also detected an increased expression of genes typical for BAT within WAT. They speculated that browning of WAT could be the compensatory mechanism preventing a further decrease in body temperature in hypothyroidism. Besides WAT browning, adipogenesis also occurs in hypothyroid mice. Endo et al. documented that not only THs but also TSH stimulates UCP1 expression in BAT of mice [19].

The results derived from animal study are in line with the findings in human clinical study of hypothyroid patients. Lapa et al. demonstrated BAT activity by visualization of 18F-FDG uptake in some subjects with thyroid cancer after total thyroidectomy [33]. Interestingly, BAT-positive patients with increased glucose utilization in BAT had higher TSH concentrations, were younger and had a lower BMI. Similar findings were presented by Broeders et al. who demonstrated 18F-FDG uptake in PET-CT in hypothyroid patients after cold exposure [34]. However, a major limitation of those studies is the small number of subjects enrolled (6 cases with BAT activation in the study by Lapa et al. and 10 patients in the study by Broeders et al.); therefore the interpretation of data needs to be drawn with caution. In a larger group of patients with subclinical or overt hypothyroidism (n = 42), it was demonstrated that cold-induced thermogenesis was decreased in hypothyroid patients and that sufficient replacement of THs restores adaptive thermogenesis [35]. In a recent study by Maushart et al., BAT activity assessed by 18F-FDG uptake in PET-CT after cold exposure was related to levels of free T4 (fT4). It was shown that cold-induced thermogenesis was approximately fourfold higher in subjects in the highest tertile of fT4 when compared to the lowest tertile of fT4 [36].

Another compensatory mechanism partially compensating for the reduction of thermogenesis could be thyroid-driven cutaneous vasoconstriction [34].

In addition to direct actions of THs on BAT, T3 may indirectly regulate thermogenesis in BAT through a central mechanism [37]. López et al. found that intracerebroventricular infusion of T3 diminished the AMPK activity in the ventromedial hypothalamus (VMH), which promotes sympathetic tone and stimulates BAT thermogenesis [38]. Central action of T3 plays a role in the upregulation of thermogenic markers, such as UCP1, hormone-sensitive lipase (HSL) and DIO2, and increases heat generation in the electron transport chain of brown adipocytes [5,6,7]. The control of BAT thermogenesis was found to be dependent on TRα1, whereas contribution of TRβ remains unknown [28]. Central actions of THs influencing BAT thermogenesis make regulatory mechanisms more complicated. A mouse model showed that THs via TRα1 signaling regulate heat dissipation through tail artery function [28].

Not only THs but also thyrotropin may be involved in the regulation of thermogenesis. TSHRs are present on brown adipocytes, and some studies suggest that TSH, by binding to TSHR, increases basal and T3-stimulated UCP1 expression [19,39]. In brown adipocytes, TSH increases DIO2 activity, lipolysis and O2 consumption, which confirm the role of TSH in thermogenesis [39]. It is proposed that TSH/TSHR signaling in BAT might be responsible for dissipating energy as heat in the process mediated by UCP1 in the hypothyroid state [5,6,19].

Another central mechanism regulating the heat production in BAT includes hypophysiotropic TRH neurons within the PVN that release TRH from axon terminals into median eminence. Once released from the hypothalamus, TRH is delivered via the hypothalamic–pituitary portal vessel system into the anterior pituitary where it stimulates secretion of TSH by thyrotrophs via TRH receptor-1. Thyrotropin stimulates T4 and T3 secretion from the thyroid. The increase of circulating THs levels after the activation of hypothalamic TRH neurons within the PVN is very likely to contribute to initiation of the thermogenic response in BAT. There is a negative correlation between serum THs concentrations and TRH expression in the PVN. The negative feedback of TRH neurons depends on THs entering through MCT8 and OATP1c1 transporters and on the interaction of β1-TR and β2-TR with T3. Tanycytes, specialized ependymal cells, play an important role in feedback control through regulating local T3 availability and by controlling TRH transport in the portal vessels [40].

Cold exposure stimulates TRH neurons in the PVN, leading to activation of the sympathetic nervous system, NE release, increased THs release and elevated activity of T3 in BAT [41,42]. Cold response within TRH neurons is transient because TRH mRNA expression is increased in the PVN within 1 h after exposure to cold and returns to original levels after 2 h. In the same experiment, T3 levels peaked after 2 h and started to return to control values [43]. The results suggest that circulating THs do not play an essential role. The fact that TRH knockout mice show cold intolerance, which is not corrected with THs supplementation, also confirms the role of TRH neurons independently of THs.

It has been shown that lean people express higher amounts of UCP1 in BAT than obese subjects [5,7]. The impaired function of BAT and the lower activity of brown fat predispose to weight gain and promote obesity. Negative relationships between BMI and the mass, as well as activity of BAT, are observed [37]. Stimulation of thermogenesis in BAT is associated with weight reduction and with numerous metabolic benefits such as increased insulin sensitivity and lipid uptake. It has been shown that white adipocytes, under specific conditions, may undergo transformation into a thermogenic phenotype. The process called “browning” or “beiging” occurs mainly in subcutaneous depots of WAT and in inguinal fat. Browning is characterized by the appearance of beige adipocytes that possess UCP1 but appear to be molecularly and functionally different from brown adipocytes. Beige adipocytes that increase in WAT are also identified as brite. It is not fully understood how beige adipocytes are formed, and the process of transdifferentiation of white to beige adipocytes is under intensive research. It has been shown that large white adipocytes can differentiate into beige adipocytes in response to cold or β3-adrenergic agonists [37]. Moreover, the beige adipocytes may lose UCP1 expression when the experimental animals are transferred back to warmer conditions, indicating that the process is reversible. It has been shown that simultaneous action of insulin and leptin on proopiomelanocortin (POMC) neurons increases the browning of white fat [44]. In addition to central mechanisms, leptin promotes browning also through peripheral mechanisms. In skeletal muscle, leptin regulates irisin expression, which can promote the browning of WAT [45]. Irisin, a myokine discovered in 2012, cleaved from its parent FNDC5 in response to exercise, seemed to be an interesting and promising factor with the browning effect. Controversy regarding irisin, its formation and function, especially in humans, remains [46]. Browning of WAT was documented after intensive exercise in mice and was attenuated in FNDC5 knockout mice [47]. WAT developed characteristics of BAT after irisin exposure in mice, and adipocytes browned when incubated with recombinant FNDC5 [48]. However, there is little knowledge about the role of irisin in humans, and the results of different studies do not confirm irisin’s response to exercise [49,50]. Currently, there is a discussion whether irisin could be a link between thyroid function and adiposity [51]. Since reliability of commercial enzyme-linked immunosorbent assays (ELISAs) for irisin measurements has been questioned [46], conclusions of such studies need to be carefully evaluated.

Another factor that may contribute to the conversion of white to brown fat tissue is fibroblast growth factor 21 (FGF21), which is produced mainly by the liver but at lower temperatures and after stimulation of β3-adrenergic receptors also by BAT [52]. FGF21 analogs and FGF21 receptor agonists constitute the group of molecules known as the FGF21 class that has beneficial metabolic effects and promotes weight and adiposity loss. FGF21 induces WAT browning; however, the mechanisms stimulating thermogenesis are not fully recognized. Several studies demonstrated that in UCP1 knockout mice FGF21 stimulates BAT thermogenesis which was UCP1-independent [53]. Those observations suggest that FGF21 is likely to stimulate BAT thermogenesis through a mechanism other than UCP1.

It is considered that some Chinese medicine agents such as resveratrol, berberine and curcumin can induce browning of WAT [54], but their detailed role in the physiology of adipose tissue needs further study.

There is evidence that THs play a role in the browning of adipose tissue by central and peripheral mechanisms (Figure 3). Alvarez-Crespo et al. showed in their mouse study that T3, acting centrally on the ventromedial nucleus of the hypothalamus (VMH), inhibits the activity of AMP-activated protein kinase (AMPK) and thus leads to increased thermogenesis in both BAT and WAT depots [55]. It has been documented that WAT browning can be blocked in a mitogen protein kinase MKK6 depending fashion and T3 participates in this phenomenon. Matesanz et al. [56] found that MKK6 deletion in mice increases T3-mediated browning, which results in MKK6-/- mice being protected from obesity induced by a high-fat diet. They also found that blocking thyroid hormone synthesis with propylthiouracil eliminates increased UCP1 expression in WAT, observed in mice lacking MKK6, while addition of exogenous T3 restores the increased energy expenditure in mice MKKP6-/-. Moreover, the researchers observed that treatment of T3 induces AMPK/p38 activation that correlates with higher levels of UCP1.To study whether the effects observed in animals are present in humans, MKKP6 expression in visceral fat of obese and lean subjects was investigated, and the analysis revealed higher levels of MKKP6 protein in obesity. In an experimental model conducted in TRβ knockout mice and UCP1 knockout mice, Johann et al. [57] documented a direct effect of T3 on browning through TRβ, regardless of the sympathetic system. In mice treated with T3, induction of hepatic expression of FGF21 via TRβ and proliferator-activated receptor α (PPARα) activation was observed [58].

In patients with obesity, the functions of WAT and BAT are significantly impaired. Proinflammatory M1 macrophages and CD4+ Th1 cells are recruited, and after infiltrating WAT and BAT depots, they block thermogenic activity. It was documented that M1-derived inflammatory cytokines inhibit UCP1 expression in BAT and WAT. Moreover, the noradrenergic signaling and catecholamine-induced lipolysis are reduced [59]. Chung et al. found that direct contact of inflammatory cells and adipocytes inhibits the recruitment of beige adipocytes [60].

## 4. Energy Expenditure and THS/Thyroid Axis

THs modulate the BMR and are important determinants for heat production during shivering and non-shivering thermogenesis [61]. THs stimulate resting energy expenditure (REE) mainly through increasing ATP production in muscles and by generating Na+/K+ as well as Ca2+ gradients. Total energy expenditure includes BMR and adaptive thermogenesis. Normal thyroid function is essential for both obligatory and adaptive thermogenesis. In humans, about 30% of energy expenditure is TH-dependent and even slight thyroid dysfunction can lead to reduction in thermogenesis and may slow down a BMR. Al-Adsani et al. [62] found that a small change of 5–10% in REE (reduction of only 75–150 kcal/day), which corresponded with a small change in TSH concentration within the normal range, could generate weight gain of several kilos in the period of 10 years. Thus, changes in body weight correlate with TSH levels. Moreover, positive associations between free T3 (fT3), fT3/fT4 ratio and visceral fat area quantified by magnetic resonance imaging (MRI) were also observed [63].

However, it is not fully known whether changes in TSH and THs levels are the cause or the consequence of differences in body weight.

## 5. Stimulation of Hypothalamus–Pituitary–Thyroid Axis in Obesity

Multiple studies on the relationship between obesity and thyroid function indicate positive correlations between BMI and TSH as well as between BMI and free or total T3. Slightly elevated TSH and T3 or fT3 concentrations are the most common abnormalities in obese children [64,65]. Some researchers consider fT4 rather than TSH to be the optimal indicator of thyroid status [66]. In obese subjects with primary subclinical hypothyroidism and with increased TSH, elevated T3 may be the result of augmented peripheral deiodination and preferential secretion of T3. It is possible that polymorphisms in DIOs converting T4 to T3 influence the proportion of THs. Moreover, leptin acts as an important regulator of peripheral and central DIOs, thereby affecting the level of THs. These mechanisms, in part, may underlie the elevated fT3 concentrations and fT3/fT4 ratios [21,63,67].

In obese individuals, elevated TSH levels may be the result of the stimulation of the HPT axis. In the animal model, a diet high in fat and simple carbohydrates led to a significant increase in T3 and TSH already in the first month of observation, and this phenomenon persisted for the next five months. At the same time, leptin receptors were significantly downregulated, and thus leptin resistance developed [68].

The stimulation of the HPT axis observed in obesity is mainly due to centrally acting leptin, which regulates the activity of neurons in the hypothalamus and has both direct and indirect effects on TRH–TSH secretion (Figure 3). TRH is a tripeptide amide (pGlu-His-ProNH2) synthesized by TRH neurons within the PVN, and then it is transported to the median eminence in the third ventricle and through the portal system to the anterior pituitary where it stimulates the secretion of TSH from thyrotrophs. TRH neurons influence energy homeostasis by central regulation of thermogenesis and, through the action of THs, stimulate mitochondrial oxygen consumption and increase thermogenesis.

There are three main groups of neurons that have synaptic connections with the hypophysiotropic TRH neurons: the arcuate nucleus (ARC) of the hypothalamus, the dorsomedial nucleus of the hypothalamus (DMN) and catecholamine-producing neurons in the brainstem [5]. The ARC contains two populations of appetite-control neurons: medial localized orexigenic neurons that synthesize neuropeptide Y (NPY) and agouti-related peptide (AGRP) and lateral anorexigenic neurons that produce POMC, α-MSH and CART. Both neuronal groups respond to peripheral nutritional signals such as changes in leptin, ghrelin, insulin and glucose.

DMN neurons also receive signals related to energy homeostasis, including leptin signaling, but these signals interact with the DMN indirectly through the ARC [69]. How DMN regulates the hypophysiotropic TRH neurons is not fully understood, but it seems to exert an inhibitory effect. Bilateral destruction of the DMN has been shown to increase the 24h release of T3, which confirms the effect on the activity of the HPT axis [70]. DMN appears to be involved in the circadian regulation of TRH hypophysiotropic neurons.

The activity of TRH neurons is also regulated by the catecholaminergic innervation. NE has been shown to stimulate TRH gene transcription. Both adrenergic and noradrenergic axons activate TRH neurons [69].

Overnutrition, through hyperleptinemia, activates TRH expression and then synthesis of thyrotropin and THs. It promotes energy expenditure by increasing BMR, thermogenesis, lipolysis and glycolysis. During fasting, the synthesis and release of TRH in the hypothalamus as well as the concentration of THs in the serum decrease.

The direct interaction of leptin on the activity of the HPT axis includes the effect on TRH-ergic neurons within the PVN through leptin receptors. The indirect effect of leptin includes regulating mechanisms of the neuronal hypothalamic network involved in the energy regulation balance. In the ARC, the critical place of the hypothalamus regulating feeding and metabolism, leptin increases the synthesis of anorexic peptides, POMC, α-melanocyte-stimulating hormone (α-MSH) and cocaine- and amphetamine-regulated transcript (CART) that are projected from the ARC to the PVN and activate pro-TRH gene expression. Moreover, leptin downregulates orexigenic peptides, NPY and AGRP, which antagonizes the α-MSH stimulatory effects on pro-TRH expression. In conditions of excessive caloric supply, leptin simultaneously suppresses appetite by increasing the expression of POMC and by inhibiting the expression of NPY and AGRP in the ARC of the hypothalamus.

However, in obesity, hyperleptinemia does not reduce appetite and does not increase energy expenditure. Those adaptive mechanisms may be disturbed if individuals were being overfed in early life stages, resulting in overeating and overweight in adults. In studies run on animals, it was found that perinatal overfeeding induced leptin resistance in the regulating neuropeptide systems of the hypothalamus. In adulthood, leptin-controlled arcuate neurons were unresponsive to signals from adipose tissue and insulin which disrupted the energy homeostasis and food intake regulation [71].

In adult Lep ob/Lep ob mice, the lack of leptin led to serious disturbances in the functioning of both AGRP/NPY and αMSH pathways [72]. However, the treatment of those mice with leptin for 20 days was unsuccessful, as it did not restore normal PVH signaling. Thus, modulation of leptin activity in the hypothalamic pathways appears to be limited to the neonatal period.

During fasting, the ARC pathway is dominant and mediates the action of leptin on TRH neurons. This thesis is based on the observation that destruction of the ARC prevents the influence of leptin on the expression of the TRH gene in the PVN [73]. Moreover, while central administration of leptin activates TRH neurons in the PVN, administration of a melanocortin antagonist completely prevents the stimulating effect of centrally administered leptin on TRH release and TSH secretion [74]. This indicates that leptin affects the hypophysiotropic TRH neurons mainly through the melanocortin system. Moreover, in NPY/MC4R double KO mice lacking signaling for both NPY and melanocortin, the fasting-induced suppression of the HPT axis is completely prevented [75].

Long-lasting excessive caloric intake and increased weight gain change the adaptive abilities of the HPT axis, and in obesity, leptin resistance appears primarily at the ARC level, but the direct effect of leptin on the PVN remains unchanged. In obese animals on a high-fat diet, POMC and MSH-α expression in the ARC remains unchanged, suggesting that increased caloric intake reduces the sensitivity of anorexigenic neurons in this nucleus to leptin [76].

## 6. Thyroid Hormones and TSH Regulate the Balance between Lipogenesis and Lipolysis and Control Adipogenesis

### 6.1. Thyroid Hormones

All TR isoforms—TRα-1, TRα-2 and TRβ-1—are present in white and brown adipocytes, and TRα-1 is more abundant than others [7]. Biological effects of THs depend on intracellular levels of T3 which is generated from T4 by the action of DIO type 1 (DIO1) and DIO2. DIO3 deactivates THs by production of reverse T3 and diiodothyronine. DIOs are selenoenzymes that are regulated by thyroid status, local selenium availability and proinflammatory cytokines [77].

DIOs are present in different tissues. DIO1 is abundantly expressed in the thyroid gland, pituitary, liver, kidneys and WAT, whereas DIO2 is present in the pituitary, muscles, placenta and BAT. Under physiological conditions, DIO3 is expressed in the brain, placenta and pancreas, while under various pathophysiological conditions, its expression begins to be present in other tissues. Little is known about DIO activity in WAT, and only DIO1 was found to be an important source of intracellular concentrations of T3. It was demonstrated that the level of DIO1 mRNA was higher in subcutaneous and visceral fat of obese when compared with non-obese subjects [78]. Moreover, DIO1 activity was elevated in obese patients, in contrast to DIO2 and DIO3 that were close to the detection limits. Moreover, there were no differences in their activities between obese vs. non-obese subjects and between different fat depots. As leptin expression was upregulated in obesity and was positively correlated with DIO1, Ortega et al. [78] speculated that leptin can modulate DIO1 activity, thereby affecting the local conversion of T4 to active T3. This phenomenon could underlie the elevated fT3 levels and higher fT3/fT4 ratios observed in obese subjects. However, dozens of controversial findings have been published in papers addressing DIO1, DIO2 and DIO3 activities in human and animal experiments and results of those studies are non-conclusive [78,79,80]. Pihlajamaki et al. analyzed transcriptomes from liver samples of obese patients, and they found that THs signaling could be suppressed, as DIO2 and DIO3 gene expression was decreased, whereas expression of DIO1 was not changed [80]. As leptin receptors were also decreased in obese individuals when compared with lean controls, it could be assumed that downregulation of leptin signaling impairs THs action in the liver and may be the cause of fat accumulation in the liver [80].

In BAT, locally produced T3 is required for thermogenic function. T3 enhances NE-induced lipolysis and heat generation. However, during cold exposure, T3 is responsible for lipogenesis, and the balance between lipolysis and lipogenesis depends on D2-mediated conversion of T4 to T3 as well as on sympathetic stimulation [81].

In WAT, T3 regulates the activity of the entire enzyme machinery responsible for lipogenesis and lipolysis. In healthy conditions, lipolysis balances lipogenesis, even if an excessive supply of energy prevails (Figure 1). T3 enhances lipolysis by increasing the number of lipolytic β2-adrenergic receptors and through postreceptor signaling, mainly c-AMP-dependent. Local production of T3, mediated by DIO2 in response to leptin, could directly affect the activity of lipolytic enzymes such as carnitine palmitoyltransferase 1α, adiponutrin, lipase of triacylglycerol, desnutrin and lipoprotein lipase. Lipolytic effect is also due to the augmentation of hormone-sensitive lipase activity, which is responsible for triacylglycerol hydrolysis in adipose tissue. The lipolytic effect of T3 is pronounced when NE is binding to adrenergic receptors [82]. In hyperthyroidism, catecholamine-induced lipolysis is enhanced and leads to a reduction of fat mass.

THs signaling is also essential for lipogenesis in both subcutaneous WAT and BAT, and β3-adrenergic stimulation plays an essential role. In vitro studies showed that insulin and T3 act synergistically on intracellular signaling pathways and induce enzymatic activity. Different studies reported that T3 stimulates the expression of lipogenic enzymes in both liver and adipose tissue: acety-CoA-carboxylase, fatty acid synthase, malic enzyme and glucose-6-phosphate dehydrogenase, especially in a feeding state high in carbohydrates and protein [83,84]. In animals on a high-carbohydrate diet, THs are known to induce lipogenesis [85].

The BAT and subcutaneous WAT are under sympathetic nervous system control, and the adrenergic stimulation plays a key role in thermogenesis, lipolysis and lipogenesis. An association between sympathetic signaling in subcutaneous WAT via the β3-AR-cAMP-PKA pathway and T3/TR signaling promoting both thermogenesis and lipogenesis was demonstrated [82]. The β3-AR agonists cannot stimulate either WAT lipogenic gene expression or UCP1 expression in the absence of THs, while T3 administration triggers these responses. In denervated adipose tissue in hypothyroid rats, lipogenesis and thermogenesis are impaired [31].

In vivo and in vitro studies have shown that T3 promotes adipogenesis in WAT through adipogenic signal pathways: C/EBPs and PPARγ (Figure 2). Impaired adipogenesis was observed in transgenic mice with mutant TRα1 [86]. Synthesis of fatty acids and simultaneously proliferation and differentiation of immature adipocytes are critical for lipid accumulation and lead to hypertrophy and hyperplasia of adipose tissue.

In addition to affecting lipid metabolism, there is evidence that THs are involved in appetite regulation. It is well established that T3 can directly increase food intake via central hypothalamic regulations (Table 1) [24].

### 6.2. TSH

TSH stimulates adipocyte lipolysis, as was proved in various in vitro experiments [20,87,88,89,90]. Lipolysis is the process of hydrolysis of triglycerides into a glycerol and FFAs that are released into circulation. The essential regulators stimulating lipolysis in humans are catecholamines (via β1, β2 and β3-AR) and natriuretic peptides, whereas the antilipolytic effect is mediated by insulin and catecholamines (through α2-AR). Several other factors such as TSH, NPY, growth hormone, glucocorticoids and tumor necrosis factor (TNF) can regulate lipolysis, either directly by specific receptors or indirectly by modulating the lipolytic cascade [90]. Stimulation of TSHR increases intracellular cAMP levels through activation of adenylyl cyclase. Perilipin and HSL are required for lipolysis and FFA release from adipocytes. It was found that thyrotropin activates phosphorylation of perilipin and HSL in a protein-kinase-A-dependent manner in mouse 3T3-L1 adipocytes as well as in primary human adipocytes [87]. Vizek et al. documented in vitro that TSH stimulated glycerol release from fat cells isolated from the subcutaneous adipose tissue of newborns but had no effect on fat cells from adults [88]. Marcus et al. found that in neonatal adipocytes, TSH could induce a significant lipolytic effect [91]. It is believed that high TSH levels occurring in infants stimulate lipolysis and FFA release that are a source of energy shortly after birth and before lactation [20,91]. This effect is reduced in the next years of life and in adults is not as significant as in newborn/pediatric age. To determine whether TSH induces lipolysis in vivo, serum FFAs were measured after administration of recombinant human TSH in patients with thyroid cancer after thyroidectomy. The lipolysis stimulation effect was evident only in subjects with BMI < 30 [87].

In vivo studies indicate a role of TSHR in the lipolytic effect of TSH. In the model of TSHR knockout mice, reduced responsiveness to the lipolytic effect of TSH and a hypertrophy of mice adipocytes were observed [92]. An increased basal lipolysis found in mice lacking functional TSHR is thought to be related to an increase in adipocyte size. Furthermore, in the study by Lundbäck et al., inactivation of TSHR resulted in the decreased expression of genes responsible for adipogenesis such as PPARγ and C/EBPs and in reduction of lipolytic adrenergic receptors including β1-AR and β3-AR [93]. Those results suggest that TSH/TSHR signaling is responsible for adrenergic activity in adipocytes. Mice lacking functional TSHR within adipose tissue exhibited the decreased expression of adiponectin gene and downregulated UCP1 in BAT. TSHR knockout animals were predisposed to obesity development.

However, the results of the above experiment are not consistent with other observations. The model of global TSHR knockout mice, adipose-specific TSHR knockout mice, thyroid-specific TSHR knockout mice and hypothyroid mice was created by Zhang et al. [94]. These researchers indicated that elevated TSH reduces energy expenditure and promotes adiposity, while mice lacking TSHR showed higher metabolic rates and were resistant to obesity. Furthermore, the browning of WAT was observed in TSHR knockout mice. In the in vitro part of the experiment, it was documented that the AMPK/PRDM16/PGC1α pathway is responsible for the thermogenic effect of TSH [94].

It is suggested that, under certain conditions, TSH may have a pro-lipogenic effect. In primary human differentiated adipocytes, it was found that insulin reduced the ability of TSH to stimulate lipolysis [89]. In the same experiment, TSH inhibited insulin signaling, and therefore it can be assumed that interactions between TSH and insulin are bilateral. It was demonstrated that lipogenesis/lipolysis balance in adipocytes depends on BMI, insulin resistance and the time of exposure to the elevated TSH concentration. At lower BMI values, the inhibitory effect of TSH on insulin-stimulated lipogenesis is dominant. However, higher BMI appeared to cause enhancement in lipogenesis, especially over a longer period of time. It seems that in adipocytes of obese subjects, when TSH and insulin act together for a long time, TSH signaling interferes with insulin signaling which leads to increased lipogenesis and reduced lipolysis [89]. In mature adipocytes, TSH can decrease expression of triglyceride lipase, which catalyzes the hydrolysis of triglycerides [95].

Thyrotropin can also increase glycerol 3-phosphate acyltransferase 3 activity. The enzyme is responsible for triglyceride synthesis and lipid storage as well as for the adipogenesis process. Ma et al. [96] found that TSH promoted fat accumulation in vivo, both by augmenting adipocytes hypertrophy and by hyperplasia of adipose tissue. In TSHR knockdown mice, the expression of genes responsible for fatty acid synthesis and triglyceride storage was downregulated and animals exhibited resistance to high-fat-diet-induced obesity. TSH stimulated lipogenesis through AMPκ/PPARγ signaling.

Different studies showed that not only THs but also TSH provides an important contribution to adipogenesis (Figure 2). Valyasevi et al. reported that orbital preadipocyte fibroblasts increased their TSHR expression with differentiation into adipocytes [17]. TSHR expression was increased at both mRNA and protein levels when 3T3-L1 preadipocytes were induced to differentiate, while knocking down TSHR blocked this stimulating effect [15]. It was showed that TSH administration in human preadipocytes with low TSHR levels did not result in significant expression of adipogenic genes and lipid accumulation whereas TSH administration in mature adipocytes with high TSHR levels increased adipogenesis-related gene expression (FASN, ADIPQ, SLC2A4) [23]. Lu et al. reported that TSHR mRNA and protein levels significantly increased during the differentiation of murine 3T3-L1 preadipocytes, and knocking down TSHR resulted in delayed cell differentiation [15]. The TSHR transcript levels in visceral fat from mice fed with a high-fat diet increased in obese mice compared to control mice fed with an ordinary diet [15]. These findings are consistent with the study by Ma et al. [96]. The ablation of TSHR in mice resulted in decreased adipogenesis and caused the resistance to high-fat-diet-induced obesity. Lu et al. [15] also analyzed the effect of obesity on TSHR transcript levels in human subcutaneous adipose tissue and they reported that it was higher in subjects with BMI > 25 than in people with BMI < 25. However, the effect of obesity on TSHR is not obvious. Nannipieri et al. [21] detected TSHR and beta-actin protein expression in subcutaneous and visceral adipose tissue obtained during abdominal surgery in 107 severely obese patients. In both adipose depots, TSHR expression was lower in obese than in 12 lean control individuals. The protein expression of TSHR in subcutaneous adipose tissue obtained from obese patients was also depressed. The findings are concordant with the observations in the animal model by Lundbäck and coworkers [93]. They found that mice with reduced TSHR expression gained weight faster than control mice. In keeping with this observation, Comas et al. [23] suggested that TSH can promote mitochondrial function in human adipocytes, thus potentially being a link between energy and lipid metabolism.

TSHR stimulation by species-specific TSH remains a topic of current investigations. In most experiments, bovine TSH is used because it effectively stimulates TSHR, especially human TSHR. It is well known that bovine TSH has a higher affinity to the human TSHR and higher signaling activity than human TSH [97]. TSH exists as a heterodimer consisting of α and β subunits, bound together to produce the active form of the hormone. The human TSH subunit β gene is located on chromosome 1, the gene coding for the α subunit is located on chromosome 6. The human TSHB gene consists of three exons, of which exon 1 is noncoding. The mouse TSH β subunit consists of five exons, and only exons 4 and 5 are the coding exons [98]. Although recombinant human TSH has a slower metabolic clearance rate and longer duration of action than does native human TSH, it is less effective. On the other hand, rat or mouse TSH is rarely available for in vivo and in vitro experiments. What is worth noting is that the observed diversity of results may be the consequence of different animal and cell models used in laboratories. Moreover, differences between species can make the interpretation of experiments difficult and confusing. Consequently, many of the reported observations of TSHR signaling and effects on extrathyroidal tissues and cells may be “non-physiological”.

### 6.3. TSH and Chronic Inflammation

Low-grade chronic inflammation, characterized by secretion of proinflammatory adipokines, leads to endothelial dysfunction, oxidative stress, atherogenesis and insulin resistance. Some authors speculate that persistently elevated thyrotropin stimulates inflammatory state [99,100,101,102,103,104,105,106,107,108], which contributes to adipose tissue dysfunction and metabolic syndrome and then may facilitate weight gain. In vitro studies revealed that TSH enhances the expression of leptin mRNA in adipose tissue and adipokine release during TSH-stimulated lipolysis [102]. In adipocytes that express TSHRs, TSH stimulates secretion not only of the leptin but also of other inflammatory cytokines, such as interleukin-6 (IL-6), TNF-α and monocyte chemoattractant protein-1 (MCP-1), which in turn lead to adipose tissue dysfunction [100,101]. In vivo and in vitro studies showed that IL-6 release from differentiated adipocytes is stimulated by TSH and the involvement of cAMP-protein kinase A pathway was suggested [99]. Significantly increased concentrations of IL-6, CRP and MCP-1 were detected in subclinical hypothyroid subjects [95,96,97,98]. In our previous study, higher TSH and IL-6 concentrations were related to metabolic syndrome [104].

It was documented that thyrotropin upregulates mRNA expression of MCP-1 and protein release from human abdominal differentiated adipocytes. Signaling through the inhibitor of the nuclear factor kappa-β kinase (IKKβ) pathway is responsible for this effect [103]. The result of another experiment documented a stimulating effect of thyrotropin on macrophages via the IκBkinase (IKKB)/nuclear factor κB pathway [100]. In the study involving autoimmune thyroiditis patients, elevated TSH levels were accompanied by increased serum concentrations of MCP-1 [107]. In patients with extreme obesity, elevated TSH concentration significantly correlated with proinflammatory cytokines such leptin, IL6, ICAM-1 and E-selectin [108]. In humans, recombinant TSH administered to patients with differentiated thyroid cancer during postoperative diagnosis stimulated leptin secretion [109].

Resistin is another adipocytokine with adverse effects on metabolism and the vascular system. In the study conducted by Eke Koyuncu et al. [110], the authors found that hypothyroid patients had significantly higher resistin serum concentrations when compared to controls. Low-grade inflammatory state is implicated with diseases such as coronary heart disease, myocardial infarction, heart failure, chronic renal disease, obstructive sleep apnea, rheumatoid arthritis, depression, colorectal, gastric and lung cancers as well as metastatic progression [111].

Not only hyperthyrotropinemia stimulates proinflammatory states, but also elevation of inflammatory cytokines may affect thyroid function [112].

### 6.4. Lipotoxicity

Obese subjects often exhibit hypertriglyceridemia which leads to accumulation of lipids in tissues other than adipose tissue. This phenomenon is known as lipotoxicity. Lipids can accumulate in the kidneys, liver, heart and skeletal muscles. Lipotoxicity has been well documented in the pathogenesis of metabolic syndrome, type 2 diabetes, nonalcoholic fatty liver disease (NAFLD) and heart failure (Figure 4).

Several studies have documented that patients with metabolic syndrome and subjects with hypertriglyceridemia have significantly higher TSH concentrations than control individuals [4,104,113,114]. Since high triglyceride concentration is often used as an indicator of the severity of lipotoxicity in human studies, one may speculate that lipotoxicity can damage thyroid function.

In the population-based case-control study conducted on 24,100 subjects, hypertriglyceridemia was associated with approximately 35% increased risk for subclinical hypothyroidism [113]. In a recent analysis of 9663 cases, Li et al. found positive correlations between TSH and triglyceride levels in patients with thyroid papillary microcarcinoma [114]. A cross-sectional study [4] of 1534 subjects found positive correlations between TSH and BMI, waist circumferences and triglyceride concentrations. In our previous study [104], the metabolic syndrome was connected with visceral fat accumulation and higher TSH concentrations. Obesity and metabolic syndrome are strongly associated with NAFLD which is characterized by enhanced hepatic lipogenesis, and it is believed that increased TSH accelerates the development of NAFLD and nonalcoholic steatohepatitis (NASH) [115]. It was documented that downregulation of the leptin pathway and decreased expression of DIO2, DIO3 and genes regulated by THs in the liver may cause fat accumulation that leads to NAFLD [80].

The results of those and various epidemiological studies support the concept that obesity through lipotoxicity may affect thyroid function and lead to the development of subclinical hypothyroidism. Concepts of high circulating triglyceride levels impacting the aggressiveness of differentiated thyroid cancer appeared [114].

The mechanism linking adiposity and thyroid function has been the subject of in vitro and in vivo experiments [113,116,117,118,119]. The direct interactions between AT and thyrocytes were studied in vitro, using a co-culture system, in which thyrocytes were cultured on the adipose-tissue fragment (ATF)-embedded collagen gel in the presence and absence of TSH [116]. Applying this method, active interaction between thyrocytes and AT was assessed. It was shown that ATF promoted cytoplasmic accumulation of lipid droplets in thyrocytes, while lipid droplets were not detected in thyrocytes without ATF. The authors suggested that lipids were transferred from ATF to thyrocytes via paracrine way. Lipid accumulation was partly dependent on TSH influence. They also found that, when thyrocytes were cultured alone, they formed flat shapes with flattened nuclei. In contrast, thyrocytes cultured with ATF formed columns with round nuclei in the bases. In addition, ATF inhibited thyrocyte apoptosis. However, this in vitro model of paracrine effects may be unable to replicate in vivo conditions. The structural and functional units of the thyroid gland are follicles that consist of thyrocytes and C cells. Follicles are supported by extracellular matrix, capillary network and stromal cells such as fibroblasts and inflammatory cells. The in vitro model by Yamamoto et al. included thyrocytes alone without follicle structures; thus this simple model does not take into consideration many complex interactions present in living organisms [116].

The effect of lipotoxicity on thyroid structure and function was evident in animals with diet-induced obesity. It was found that the thyroid glands of rats fed with a high-fat diet for 12 weeks displayed deleterious structural and functional changes. Microscopically large follicles with excessive amounts of colloid were observed. In addition, capillaries between increased follicles were compressed. High-fat diets provoked the infiltration of inter-follicular connective tissue and connective tissue surrounding the thyroid gland by mast cells. It is known that, in obesity, the mast cells secrete different proinflammatory cytokines and chemokines that stimulate a state of tissue inflammation. Thus, in obesity induced by a high-fat diet, structural changes occur that may in turn lead to worsened thyroid function [117].

Diet-induced obese mice showed increased expression of lipogenesis-regulation genes, such as sterol regulatory element binding protein 1 (SREBP-1), peroxisome proliferator-activated receptor γ (PPARγ), acetyl coenzyme A carboxylase (ACC) and fatty acid synthetase (FASN) in the thyroid gland. De novo lipogenesis in thyrocytes triggered defects in thyroglobulin synthesis and led to thyroid dysfunction. After 8 weeks of the experiment, serum leptin in those mice was markedly elevated with a simultaneous increase in TSH and a decrease in THs concentrations [118].

A study by Shao et al. showed that long-term high-fat feeding induced lipotoxicity in the thyroid glands in rats [119]. The excess of cellular lipids induced endoplasmic reticulum (ER) stress in thyrocytes and changed the structure of the thyroid gland and the serum THs levels. The authors also observed that after a 24-week withdrawal of a high-fat diet the above lesions did not improve, which suggests that the duration of the harmful diet is important.

It was found that ER stress decreased the expression of genes responsible for TH synthesis in thyrocytes [120]. Under ER stress, the expression of TSHR and signaling of TSH/TSHR were reduced.

Zhao et al. reported that diet-induced lipid accumulation in human thyrocytes downregulates the expression of sodium iodide symporter, thyroglobulin and thyroperoxidase [121]. Those molecules are involved in the THs synthesis, and their suppression may affect thyroid function and can be related to subclinical hypothyroidism. In a study involving obese human participants, the thyroid tissue obtained from lobes removed during thyroidectomy was characterized by expansion of interfollicular fat depots and thyroid steatosis [118].

Typical lipomatous lesions of the thyroid gland characterized by diffuse stromal infiltration by mature adipocytes within the thyroid is a rare condition [122]. Thyroid lesions containing excess fat include heterotopic fat nests, thyrolipoma, liposarcoma and thyroid papillary or follicular cancer [123]. Schroder et al. suggested that adipose tissue deposited within thyroid parenchyma may result from metaplasia of stromal fibroblasts due to impaired circulation or tissue hypoxia [122].

## 7. Interactions between Adipocytokines and Thyroid Hormones/TSH

WAT secretes a number of adipocytokines that influence the target tissues such as the liver, skeletal muscle, central nervous system, heart, vascular system and pancreas. Besides their endocrine action, adipokines exert autocrine and paracrine impact on adipocytes. Similarly to WAT, BAT also synthesizes “batokines” such as FGF21, neuregulin 4, and VEGF and cytokines such as IL-6.

Fat tissue can affect thyroid function via adipocytokines, mainly through leptin, which controls TSH secretion. Leptin exerts the central role in regulating the HPT axis and by activation of TRH neurons leads to increased production of THs. Interactions between the HPT axis and leptin are bilateral; it was reported that administration of recombinant TSH induces a rise in leptin in proportion to adipose mass [108].

Significant positive correlations between leptin and TSH were observed in many studies [124,125,126]. A positive correlation between TSH and leptin/adiponectin ratio was recently reported [127]. Leptin/adiponectin ratio is considered to be a good marker of adipose tissue dysfunction, thus van Tienhoven-Wind et al.’s observations suggest a relationship between the HPT axis and impaired adipose tissue function.

Recently, it has been reported that leptin plays a role in thyroid cancerogenesis bystimulating tumor cell growth and invasion [128,129]. Warakomski et al. [129] documented that higher leptin levels were connected with more advanced clinical stages of well-differentiated thyroid cancer.

FGF21, produced mainly by the liver, under T3 control (Figure 4), is another link between adipose tissue and the HPT axis [130] (Figure 4). Many studies have shown FGF21 expression in visceral, subcutaneous, epicardial and cervical white and brown adipose tissue, but some studies conducted in human adipose tissue reported undetectable FGF21 gene expression [130]. It was documented that FGF21 becomes elevated as obesity develops, and positive correlations between BMI and FGF21 were observed [131]. In the liver, FGF21 increases β-oxidation of fatty acids and downregulates lipogenesis. In WAT, FGF21 stimulates lipolysis and the browning process. However, in contrast to the induction seen in the liver, Adams et al. [58] found reduced expression of FGF21 in WAT in response to T3 administration. Although the beneficial metabolic effects of FGF21 are well documented, the signaling pathways remain unresolved.

The results of studies of interactions between thyroid state and adiponectin, vaspin and visfatin are conflicting [132]; therefore their roles in HPT axis function are not well characterized, and this issue requires further studies.

Recently, the effects of leptin and adiponectin on the morphology and thyroglobulin expression of thyrocytes alone in the absence and in the presence of TSH were assessed in an in vitro experiment [116]. Leptin increased cellular hypertrophy of thyroid cells and enhanced thyroglobulin expression. Low leptin concentrations had no effect on thyrocytes, whereas higher concentrations resulted in hypertrophy up to a certain limit, above which they had no effect. It is evident that the above phenomenon is dependent on the appropriate concentration of leptin. In contrast, adiponectin did not affect the morphology of thyrocytes and thyroglobulin expression. The stimulating effect of leptin was enhanced by administering it together with TSH. This in vitro model does not replicate in vivo conditions; therefore further in vitro and in vivo studies are required to evaluate precisely the mechanism of interactions between adipocytokines and the thyroid.

## 8. Conclusions

The purpose of this review was to summarize available data characterizing the complex relationships between adiposity and thyroid function. TSH and THs are important regulators of food intake and energy storage. They play regulatory roles in thermogenesis, lipogenesis/lipolysis balance and adipogenesis. Elevated TSH may trigger low-grade chronic inflammation. It is known that little variations in thyroid function are closely related to weight changes and adipose tissue can affect thyroid function through different mechanisms. Lipotoxicity, chronic inflammation and adipocytokine release play a role. In humans, even a slow thyroid dysfunction may contribute to the gain of body and fat mass, and in obese subjects, slightly elevated TSH may be the result of subclinical hypothyroidism. Finding the cause of hypothyroidism is essential in the interpretation of TSH tests. However, it can be assumed that hyperthyrotropinemia may be the result of activation of the hypothalamic–pituitary–thyroid axis. A measurement of fT3 concentration may help in differentiating causes of the elevated TSH levels. It is speculated that obesity, inflammatory state and lipotoxicity may lead to dysfunction of the thyroid gland; therefore, promoting weight loss through dietary and lifestyle intervention is necessary to slow down unfavorable phenomena.

## Figures and Tables

**Figure 1 ijerph-18-09434-f001:**
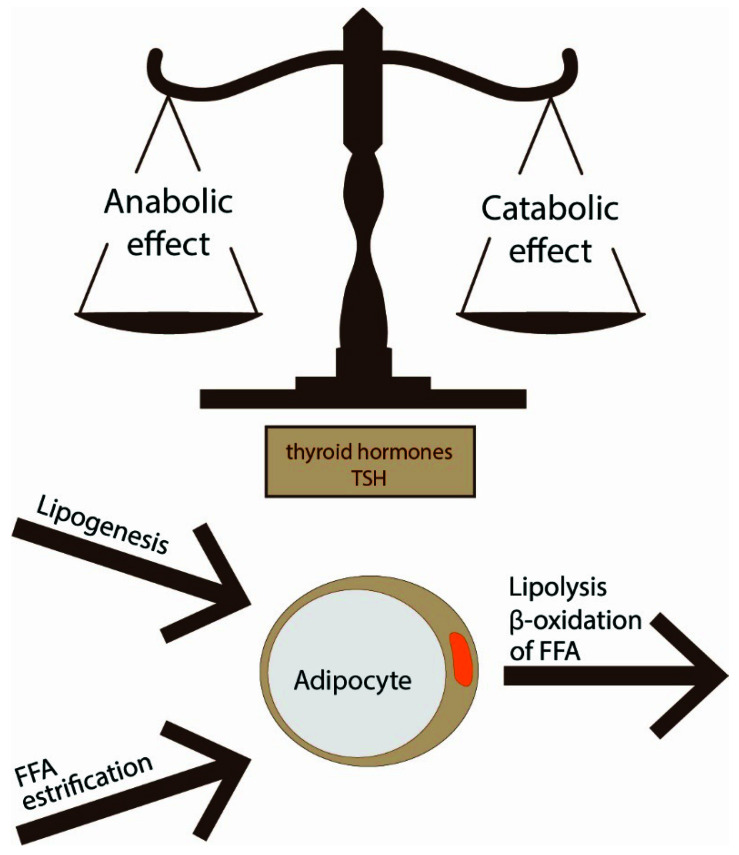
The size of adipocytes is the result of the balance between lipogenesis, lipolysis and fatty acid oxidation; these processes are regulated by THs and TSH.

**Figure 2 ijerph-18-09434-f002:**
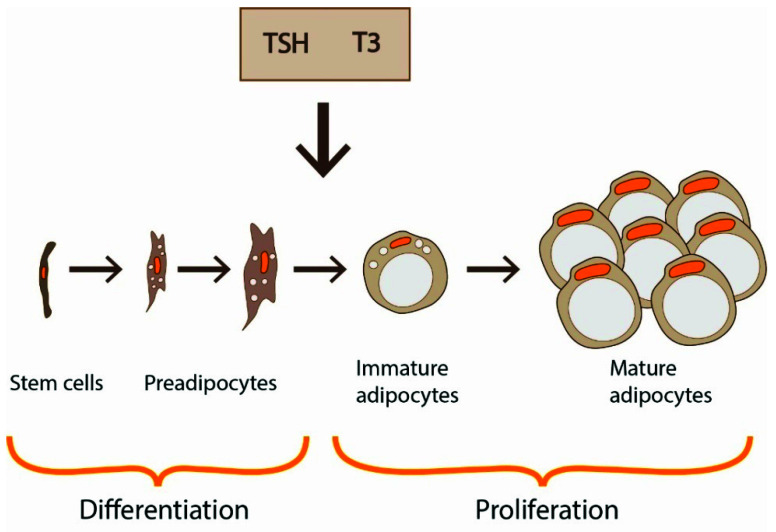
Adipose tissue grows by two mechanisms: hyperplasia (increased number of fat cells) and hypertrophy (increased adipocyte volume). The differentiation of progenitor cells through preadipocytes into adipocytes, as well as the proliferation of adipocytes, is also under TSH/THs control.

**Figure 3 ijerph-18-09434-f003:**
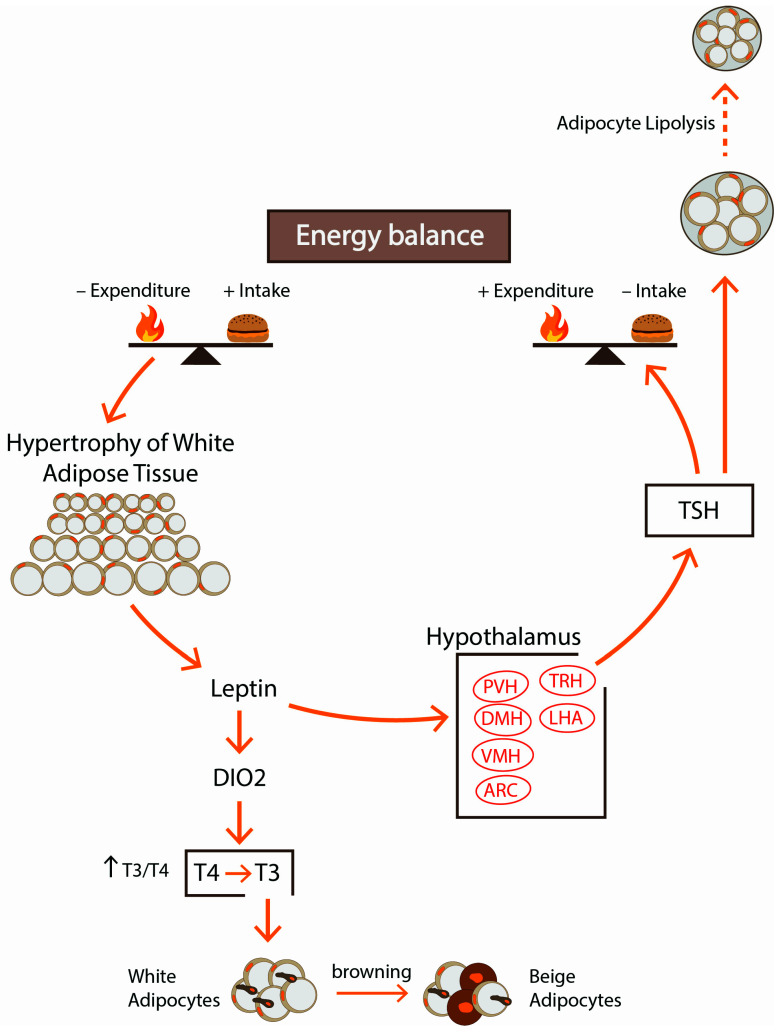
In conditions of excessive caloric supply, leptin activates TRH expression and then synthesis of TSH and THs. It suppresses appetite and promotes energy expenditure by increasing lipolysis and thermogenesis. In obese people, leptin can activate DIO expression and can increase conversion of T4 to T3. THs play a role in the browning of adipose tissue.

**Figure 4 ijerph-18-09434-f004:**
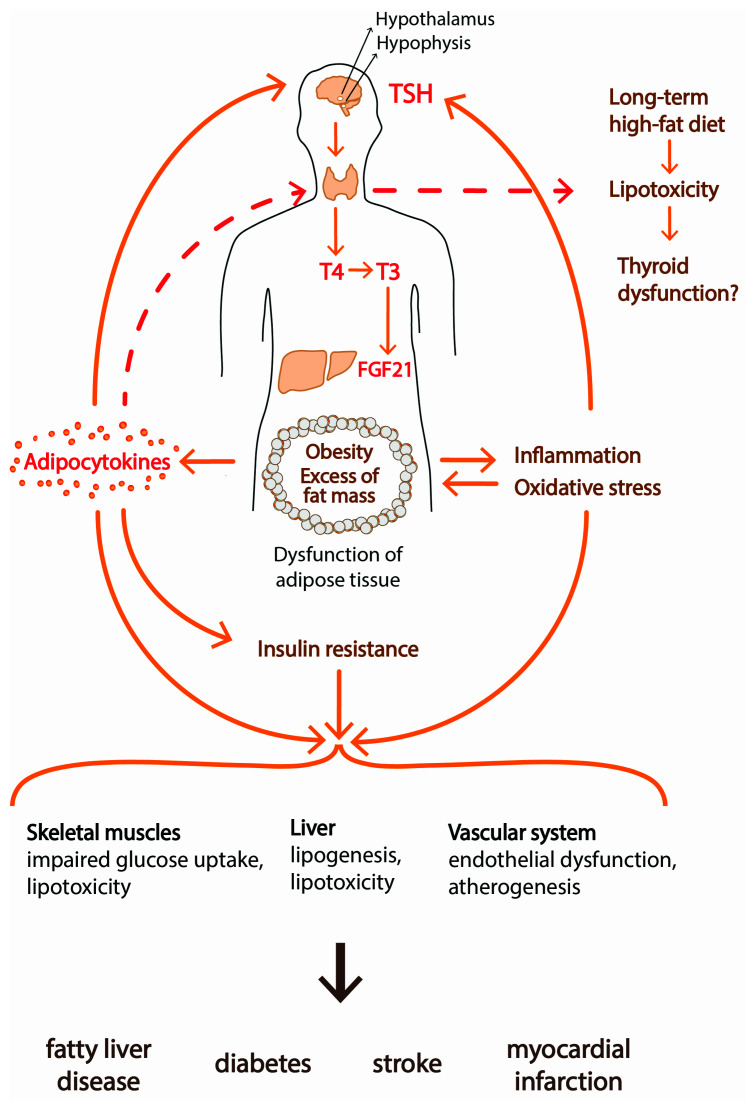
Adipose tissue dysfunction through insulin resistance, altered secretion of adipocytokines and inflammatory state are involved in the development of diabetes mellitus and vascular diseases. The excess of adipose tissue may initiate the lipotoxicity and inflammatory process within the thyroid gland. That leads to the dysfunction of the thyroid gland and results in an energy imbalance. FGF21 is produced mainly by the liver, under T3 control. In the liver, FGF21 increases β-oxidation of fatty acids and downregulates lipogenesis. In WAT, FGF21 stimulates lipolysis and the browning process.

**Table 1 ijerph-18-09434-t001:** Effect of TRH, TSH and T3 on appetite.

Hormone	Food Intake
TRH	↓
TSH	↓
T3	↑

## Data Availability

Not applicable.

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
