# Peer review of "Obesity and Thyroid Axis"

_ijerph, 2021, doi:10.3390/ijerph18189434_

Round 1
Reviewer 1 Report
Authors present a comprehensive review on recent aspects of the relationships between obesity and thyroid hormone (TH) status. However, it is not evident which new ideas or concepts are intended to be communicated beyond what has been published reviewed and discussed before.
General comments:
Anything new in this review which has not been presented and written several times before?
Authors list a large number of observation of associations and correlations between TSH / TH and obesity or metabolic parameters related to obesity. What is the scientific or clinical goal of this descriptive report about previously published reports?
An encyclopedic attempt covering what has been reported during the decade or the last few years?
There is also a lack of critical discussion of the contented repeated and summarized, in my opinion a major task to be requested for a review on a very popular subject like this, where not all available data are in agreement (see WAT, BAT, beige controversies, thermoneutrality issues, central vs peripheral regulation, neuronal vs endocrine inputs, etc.).
Obviously, obesity is as a highly important health issue, quite similar to the pandemic thyroid-related diseases in the past and the present. So, what is intended here?
The majority of references in the first part of this review refer to another previous reviews which is not very helpful for the readers.
Detailed comments:
Introduction
l. 28 ff: Too narrow view of T3 and TR action. Evidence for non-canonical mechanisms of TH action, interference with mitochondrial function (3,5-T2) and direct TH effects via plasma membrane and non-nuclear targets and processes.
l. 34 ff, and main text: It needs to be discussed in more detail what the evidence for functional signalling by TSH receptor really is in preadipocytes, white, beige and brown adipocytes. TSHr transcript expression is not sufficient, in vitro data are not sufficient, any knock down or k.o. data from in vivo observations?
Obesity or changes in adipocyte function and adipose tissue characteristics, distribution etc. in patients with TSHr stimulating antibodies? The three quoted references do not support this idea and earlier papers proposing this concept are not quoted (see e.g. refs in doi: 10.1038/s41366-018-0203-1; doi: 10.1186/1476-511X-11-17).
Considering the continuing controversy of functional TSH receptors in non-thyroidal tissues and their specific stimulation by species-homologous TSH (and not by any other (species) glyco-proteo-hormones), a more precise presentation and discussion of this hypothesis is required.
2.
l. 50: for every three FFA also one glycerol molecule is mobilized, which should not be forgotten considering its role in the CHO metabolism.
3.Thermogenesis:
l. 72 ff: Discussion and inclusion of a recent fresh look at the relationships between TH and thermogenesis might markedly improve this review, e.g. doi: 10.1530/EC-20-0562.
l. 99: Do hypothyroid patients with elevated TSH show elevated thermogenesis? Probably just the opposite is the case considering typical clinical features of hypothyroid patients (and rodent animal models).
l. 124: There is a major ongoing controversy on irisin, its formation and function especially in humans. This needs to be mentioned and discussed here, before far reaching speculations are made. see e.g.: Irisin: Still chasing shadows doi: 10.1016/j.molmet.2020.01.016.
l. 128: Also, for FGF21 various species differences have been reported in terms of its production, function and regulation. Thus, general statements as written here are not very helpful and must be more precisely related to the species context.
l. 141 ff: This whole chapter requires specification on which effects are observed in humans or animal experimental models.
4.
l. 169: The term “normal range” should not be used in 2021 anymore considering the high variability of human phenotypes in the era of personalized precision medicine; must be replaced by REFERENCE RANGE!
l. 170: Authors need to restrict this statement to adults, considering that in children and adolescents somewhat different relationships have been reported.
5.
l. 181: the term “The enhancement of the HPT axis” makes not much sense.
l. 190 ff, throughout text: Please do not create an unusual new abbreviation, use ARC.
6.1
l. 250 ff: Statements in this paragraph need to be supported by appropriate references.
l. 255: not clear whether Dio1 activity changes under these conditions are cause or effect of altered circulating T3!
Expression of DIO2 and DIO3 is decreased in liver of obese subjects. DIO1 activity is not significantly altered in obese vs. lean humans (doi: 10.1210/jc.2009-0212).
Again, species issues: in mice: changes in D1 activity in white adipose tissue under the conditions of changing adiposity, and a stimulatory effect of leptin on D1 activity in the tissue (doi: 10.33549/physiolres.931866), in Araujo rats: augmented deiodinase type 1 activity in liver and kidney and increased serum rT3 in HF rats (doi: 10.1210/en.2010-0026)
Highly controversial findings have been published in a dozen of papers addressing this issue of Dio1, Dio2 and Dio3 activities in humans and several animal exp models (see doi: 10.1038/ijo.2).
6.2
l. 299 ff: This statement is more than misleading because the quoted references do not relate to in vivo but merely to in vitro studies! Authors need to be precise in their descriptions and interpretation.
Elevated TSH is known to stimulate TH secretion in vivo in intact organisms, thus effects incorrectly attributed to TSH elevation might be exerted via resulting systemic or local T3 action!
Please distinguish between these situations as long as no clear and unequivocal demonstration of functional TSHr and their signalling cascade has been made for various adipocyte cell types under in vivo experimental conditions.
TSH action and TSHr function in extrathyroidal tissues is still highly controversial and data using tissue specific TSHr ko/kd models are not yet clear.
Which of these studies were performed with pure recombinant species-specific TSH, which with commercial TSH containing other (glycoproteohormone) contaminations?
In which of these (in vitro) studies were the endogenous culture medium TH concentration controlled and reported and were the results different then depending on TH medium concentrations?
l. 330: ref 53 refers to mouse ESC, mistake?
l. 332: It must be clearly stated that authors only analyzed transcript levels and no TSHr protein or function.
6.3
l. 336: Again a statement exaggerating the underlying publications: The two quoted papers come from only one group and the experiments are simply in vitro observations and too far reaching speculations related to this are not justified!
l. 344: Again references 56-58 are only referring to in vitro data and only ref 59 reports on observations in patients. These two things need to be separated and more clearly reported here.
l. 373: Authors might consult a classical paper on this topic and the reconsider what and how they report and discuss this postulated link between thyrocytes and 'glandular' adipocytes: PMID: 3842076.
6.4
l. 408: NIS does not transport the element iodine but solely its anion and is thus called iodide symporter!
7.
l. 427 ff: Hasn't this already been stated before in this review? Why repeated here?
l. 435 ff: Isn't that all an observation related to this highly artificial in vitro experiment co-culturing adipose tissue (chunks) together with primary culture human thyrocytes on a collagen layer leading to over-interpretation beyond the experimental limits of this model ?
Figures are instructive and clear.
Minor points:
l. 135: evidence has no plural!
l. 158: first word in this line does not exist in English.
Author Response
Dziękujemy recenzentowi za komentarze. Jesteśmy wdzięczni za zwrócenie uwagę na błędy, które oczywiście zostały poprawione. Mamy nadzieję, że zajęliśmy się wszystkimi podniesionymi pytaniami, a zmodyfikowany dokument zostanie przyjęty do publikacji w International Journal of Environmental Research and Public Health

Reviewer 2 Report
I have read with interest this interesting review. I have made several minor suggestions in a pdf with sticky notes.
One major comment I had in section 5, was that ....significant increase in T3 and TSH indicates primary subclinical hypothyroidism where preferential secretion of T3 should be expected (Reference: https://doi.org/10.1016/j.ejpn.2007.03.002)
I would expect as a clinician rather than a basic researcher to see more straightforward conclusions related to the clinical significance of evaluating the thyroid axis in obesity and perhaps through those to hear the authors' opinion about the egg-chicken puzzle: Should we seek normalizing or even optimizing the thyroid axis in obesity perhaps after having previously tried to reinstate it by weight loss through lifestyle modification strategies? I am just posing a question and I would like very much to hear the authors' opinion on that. Maybe a small paragraph deserves a place in this otherwise well-written review to provide a useful clinical message from the accumulated knowledge of the authors on the subject.

Author Response
We thank the reviewer for the comments. We are grateful for pointing out the errors, which of course have been corrected. We hope we addressed all the questions raised and the modified paper will be accepted for publication in International Journal of Environmental Research and Public Health

Round 2
Reviewer 1 Report
see attached pdf!

Author Response
We thank the reviewer for the comments.
We carefully read and analyzed the review of our paper. We agree with all comments and revised the paper accordingly.
Below there is a list of all changes made.
Major points Rebuttal letter point 6.1:
Authors need to precisely describe that this data is only for transcripts, while no corresponding activities for DIO2 or Dio3 were demonstrated. The DIO1 is the DIO isoenzyme preferentially expressed in liver, while rarely any DIO2 nor any DIO3 enzyme activity have been reported in human (and rodent) liver. Transcript data of selenocysteine encoding genes have to be interpreted with great caution due to the posttranscriptional cotranslational selenocysteine incorporation mechanism, strongly regulated by the local selenium availability, inhibited by proinflammatory cytokine and other factors. Thus, as long as no enzyme activity for selenoprotein enzymes nor their protein expression (immunohistochemistry, western blot etc.) has been documented, transcript levels might be meaningless and without biological relevance because many of these transcripts undergo nonsense mediated decay typical for such sleenoprotein transcripts! If authors really would have read ref 73 fully they would have seen that neither DIO2 nor DIO3 activity was found in contrast to DIO1 activity! Thus, their interpretation of this data is misleading in suggesting to cursory readers that DIO2 and DIO3 activities might have been measured and found decreased like the transcripts analyzed!
OUR ANSWER: Following the reviewer`s advice, we have modified the mentioned above text as follows:
Old version: DIOs activity has been investigated in animals and human samples collected durning biopsies. Results of those studies are non conclusive and further investigations are necessary [71]. DIO1 is abundantly expressed in the thyroid gland, pituitary, liver, kidneys and WAT, whereas DIO2 is richly present in pituitary, muscles, placenta and BAT. In obese people, leptin activates DIOs expression and can mediate in enhanced local T3 production in fat depots. Ortega et al. demonstrated increased expression and activity of DIO1 in adipose tissue of obese subjects [71]. Moreover, positive correlations between DIO1 and leptin expression was found. The researchers conluded that local production of T3, mediated by DIO1 in response to leptin, may affect adipose tissue metabolism. This phenomenon could also underlie the higher fT3 and fT3/fT4 ratios in obese subjects than in controls. However, dozens of controversial findings have been published in papers addressing DIO1, DIO2 and DIO3 activities in humans and several animal experiments [71-73]. In human liver samples collected during liver biopsies from obese and lean individuals, it was found that expression of DIO2, DIO3 and leptin receptors were decreased in obese subjects when compared with lean controls, whereas the DIO1 activity was not significantly altered. It is generally assumed that down regulation of leptin signaling impairs thyroid hormones actions in the liver and may be the cause of fat accumulation in the liver of obesity [73].
New version is as below:
„DIOs are selenoenzymes that are regulated by thyroid status, local selenium availability and proinflammatory cytokines [77]. DIOs are present in different tissues. DIO1 is abundantly expressed in the thyroid gland, pituitary, liver, kidneys and WAT, whereas DIO2 is present in pituitary, muscles, placenta and BAT. Under physiological conditions, DIO3 is expressed in the brain, placenta and pancreas, while under various pathophysiological conditions, its expression begins to be present in other tissues. Little is known about DIOs activity in WAT, and only DIO1 was found to be important source of intracellular concentrations of T3. It was demonstrated that DIO1 mRNA was higher in subcutaneous and visceral fat of obese when compared with non-obese subjects [78]. Moreover, DIO1 activity was elevated in obese patients, in contrast to DIO2 and DIO3 that were close to the detection limits. Moreover, there were no differences in their activites between obese vs non-obese subjects, and between different fat depots. As leptin expression was upregulated in obesity and was positively correlated with DIO1, Ortega et al. [78] speculated that leptin can modulate DIO1 activity, thereby affecting the local conversion of T4 to active T3. This phenomenon could underlie the elevated fT3 levels and higher fT3/fT4 ratios observed in obese subjects. However, dozens of controversial findings have been published in papers addressing DIO1, DIO2 and DIO3 activities in humans and animal experiments and results of those studies are non conclusive [78-80]. Pihlajamaki et al. analyzed transcriptomes from liver samples of obese patients and they found that THs signaling could be suppressed, as DIO2 and DIO3 gene expression was decreased, whereas expression of DIO1 was not changed [80]. As leptin receptors were also decreased in obese individuals when compared with lean controls, it could be assumed that down regulation of leptin signaling impairs THs actions in the liver and may be the cause of fat accumulation in the liver [80].”
- 34 ff: authors did not respond clearly to the question: “…any knock down or k.o. data from in vivo observations?…”
OUR ANSWER: In various sections of the manuscript we have quoted publications on knockdown and knockout mice. See below:
“Lu et al. reported that TSHR mRNA and protein levels significantly increased during the differentiation of murine 3T3-L1 preadipocytes and knocking down TSHR resulted in delayed cell differentiation [15].”
“While TRα1 is necessary for the proper adrenergic stimulation of brown adipocytes, TRβ isoform is responsible for T3-regulated UCP1 activation that was confirmed in TRβ gene knockout mice [27,28].”
“Browning of WAT was documented after intensive exercise in mice and was attenuated in FNDC5 knockout mice [47].”
“In the model of TSHR knockout mice a reduced responsiveness to lipolytic effect of TSH and a hypertrophy of mice adipocytes were observed [92].”
“Furthermore, in the study by Lundbäck et al., inactivation of TSHR resulted in the decreased expression of genes responsible for adipogenesis such as PPARγ, C/EBPs, and in reduction of lipolytic adrenergic receptors including β1-AR and β3-AR [93].”
“The model of global TSHR knockout mice, adipose-specific TSHR knockout mice, thyroid-specific TSHR knockout mice and hypothyroid mice was created by Zhang et al. [94].”
“Ma et al. [96] found that TSH promoted fat accumulation in vivo, both by augmenting adipocytes hypertrophy and by hyperplasia of adipose tissue. In TSHR knockdown mice, the expression of genes responsible for fatty acid synthesis and triglycerides storage was downregulated and animals exhibited resistance to high fat diet induced obesity.”
In their revised text to this issue author elaborate on TSHb:
What about expression of TSH alpha subunit in these tissues and experimental evidence? TSHb alone cannot activate the TSHr, which is only responsive to the intact TSHab dimer and TSHr binding auto-AB. So again what is the meaning of a subunit transcript altered. Has the translated TSHb protein been shown in these tissues? Has the functional protein dimer been identified in this tissue? The reviewers comment on species-specific TSH issues: Probably the reviewer was not clear in his comment: There is an ongoing debate on TSHr stimulation by species specific TSH. Typically, in most experiments bovine TSH is and has been used which is very potent in stimulating TSHr of most species, especially also human TSHr. However, rh TSH shows different (less effective) stimulation, and rat or mouse TSH is rarely available for in vivo and in vitro exp. using these experimental models and cells/cell lines derived therefrom. The downstream signalling of TSHr via Gs, Gq11, Gi etc. differs not only in various species but also between tissues and cell lines. Thus, many of the reported observations on claimed TSHr signalling and effects in extrathyroidal tissues and cells might be "unphysiological".
Authors should be more careful and critical in their discussion and interpretation of those observations on biological effects (in vivo and in vitro) of various TSH preparations.
OUR ANSWER: We are grateful for this comment. The following text has been added:
„TSHR stimulation by species specific TSH remains a topic of current investigations. In most experiments bovine TSH is used because it effectively stimulates TSHR, especially human TSHR. It is well known that bovine TSH has a higher affinity to the human TSHR and higher signaling activity than human TSH [97]. TSH exists as heterodimer consisting of α and β subunits, bound together to produce the active form of hormone. The human TSH subunit β gene is located on chromosome 1, the gene coding for the α subunit is located on chromosome 6. The human TSHB gene consists of 3 exons, of which exon 1 is noncoding. The mouse TSH β subunit consists of 5 exons and only exons 4 and 5 are the coding exons [98]. Although recombinant human TSH has a slower metabolic clearance rate and longer duration of action than does native human TSH, is less effective. On the other hand, rat or mouse TSH is rarely available for in vivo and in vitro experiments. What is worth noting, observed diversity of results may be the consequence of different animal and cells models used in laboratories. Moreover, differences between species can make the interpretation of experiments difficult and confusing. Consequently, many of the reported observations of TSHR signaling and effects on extrathyroidal tissues and cells may be "non-physiological".”
Answer to Qu l. 141 ff: The first paragraph of this new “There is evidence that thyroid…” text does not give any info that these are mouse studies in contrast to the authors rebuttal statement!
OUR ANSWER: We have overlooked this mistake, the text has been corrected, see below:
„There is evidence that THs play a role in the browning of adipose tissue by central and peripheral mechanisms (Figure 3). Alvarez-Crespo et al. have shown in the mouse study that T3, acting centrally on the ventromedial nucleus of the hypothalamus (VMH), inhibits the activity of AMP-activated protein kinase (AMPK), and thus leads to increased thermogenesis in both BAT and WAT depots [55].”
Revised ms text: l.22: is this really the case? Any support by a major clinical guideline? Any support by a major review?
OUR ANSWER:
1/In 2020 European Society of Endocrinology recommended TSH measurements in obesity
[1] Pasquali, R.; Casanueva, F.; Haluzik, M.; van Hulsteijn, L.; Ledoux, S.; Monteiro, M.P.; Salvador, J.; Santini, F.; Toplak, H.; Dekkers, O.M. European Society of Endocrinology Clinical Practice Guideline: Endocrine work-up in obesity. Eur. J. Endocrinol. 2020, 182, 1-32, doi: 10.1530/EJE-19-0893.
2/A study conducted on 4649 subjects found the relationship between TSH and the BMI index.
[2] Knudsen, N.; Laurberg, P.; Rasmussen, L.B.; Bülow, I.; Perrild, H.; Ovesen, L.; Jørgensen, T. Small differences in thyroid function may be important for body mass index and the occurrence of obesity in the population. J. Clin. Endocrinol. Metab. 2005, 90, 4019-4024, doi: 10.1210/jc.2004-2225.
We also cited a study on 2818 persons which showed TSH increasing with BMI
[3] Abdi, H.; Faam, B.; Gharibzadeh, S.; Mehran, L.; Tohidi, M.; Azizi, F.; Amouzegar, A. Determination of age and sex specific TSH and FT4 reference limits in overweight and obese individuals in an iodine-replete region: Tehran Thyroid Study (TTS). Endocr. Res. 2021, 46, 37-43, doi: 10.1080/07435800.2020.1854778.
and a study on 1534 adults that showed that TSH within the reference range was positively related with the prevalence of overweight/obesity
[4] Lai, Y.; Wang, J.; Jiang F.; Wang, B.; Chen, Y.; Li, M.; Liu, H.; Li, C.; Xue, H.; Li, N.; Yu, J.; Shi, L.; Bai, X.; Hou, X.; Zhu, L.; Lu, L.; Wang, S.; Xing, Q.; Teng, X.; Teng, W.; Shan, Z. The relationship between serum thyrotropin and components of metabolic syndrome. Endocr. J. 2011, 58, 23-30, doi: 10.1507/endocrj.k10e-272.
See the text below:
„Thyroid hormonal tests are routinely ordered when seeking obesity origins [1] . Often higher levels of serum thyrotropin (thyroid stimulating hormone, TSH) within the reference range or slightly elevated levels are found in the obese state [2-4].”
- 106 ff: Was TSH b protein detected too? If not how would TSHb alone act as it is known to be not sufficient to activate its receptor? So what does the rh TSH effect statement really mean?
(see also above, comment to rebuttal letter).
OUR ANSWER:
The old version TSH consists of two subunits, a common α subunit and a unique β subunit. Recently, Comas et coworkers found the expression of TSH β gene expression in both visceral and subcutaneous adipose tissue, however, its specific role is uknown [19]. TSH β mRNA was positively correlated with the expression of mitochondrial function and fatty acid mobilization-related genes expression that suggests the involvement in energy expenditure. Interestingly, administration of recombinant TSH resulted in increased mitochondrial respiratory capacity and ATP production.
was changed and the following explanation was added:
“TSH consists of two subunits, a common α subunit and a unique β subunit. Recently, TSH β (TSHB) gene mRNA and protein expression in both visceral and subcutaneous human adipose tissue were described, however, TSH alpha subunit mRNA and protein were not demonstrated [22,23]. TSHB mRNA was positively correlated with the expression of mitochondrial function and fatty acid mobilization-related genes that suggests the involvement in energy expenditure [23]. Adipose tissue TSHB protein level was positively correlated with mitochondrial respiratory capacity only in visceral depot. Interestingly, no association was found between TSH protein and TSHB gene expression [23]. Moreover, adipose tissue TSHB mRNA was significantly correlated with serum total and LDL-cholesterol levels in 3 independent groups of patients (cohort 1 = 38 patients with mean BMI 45.1±6.4 kg/m2, cohort 2 = 73 subjects with BMI between 19.5-85.2 kg/m2, cohort 3 = 20 participants with mean BMI 42.8±5.5 kg/m2). The results provided evidence that TSHB is implicated with cholesterol metabolism in human adipose tissue. Both, visceral and subcutaneous adipose tissue TSHB correlated positively with mRNA of 3-hydroxy-3-methylglutaryl-CoA reductase (HMGCR), the key enzyme in cholesterol biosynthesis. No associations were found between serum total and LDL cholesterol and TSHR expression in adipose tissue [22]. Administration of recombinant human TSH resulted in increased HMGCR mRNA levels in fully differentiated adipocytes, but not in preadipocytes [22]. Treatment of human mature adipocytes with recombinant TSH enhanced mitochondrial respiratory capacity and ATP production, as well as led to increased adipogenic-related genes, but no effects were observed in human preadipocytes with low expression of TSHR [23]. As inverse correlation between thyroxine and TSHB mRNA in adipose tissue was found, Moreno-Navarrete et al. suggested that local TSH production may be related to THs [22]. Moreover, researchers demonstrated that the main source of TSHB was stromal vascular cell fraction. All the observations make the detailed role of TSHB in adipose tissue not fully understood but it seems likely that TSHB acts as a paracrine factor that is implicated in lipids metabolism and energy homeostasis. In order to explore specific mechanisms linking TSHB and adipocyte function, further studies are needed.”
- 188 ff: Sentence unclear or incomplete
OUR ANSWER: We could not find the unfinished sentence. We read the paper carefully several times.
l 249 ff paragraph: Several controversial papers appeared recently on this topic.
OUR ANSWER: We changed the old version:
The results derived from animal study are in line with the findings in human clinical study of hypothyroid patients. Lapa et al. demonstrated BAT activity by visualization of 18F-FDG uptake in some subjects with thyroid cancer after total thyroidectomy. Interestingly, BAT-positive patients with increased glucose utilization in BAT had higher TSH concentrations, were younger and had lower BMI [29]. Similar findings were presented by Broeders et al. who demonstrated 18F-FDG uptake in PET-CT in hypothyroid patients after cold exposure [30].
into following text:
„The results derived from animal study are in line with the findings in human clinical study of hypothyroid patients. Lapa et al. demonstrated BAT activity by visualization of 18F-FDG uptake in some subjects with thyroid cancer after total thyroidectomy [33]. Interestingly, BAT-positive patients with increased glucose utilization in BAT had higher TSH concentrations, were younger and had lower BMI. Similar findings were presented by Broeders et al. who demonstrated 18F-FDG uptake in PET-CT in hypothyroid patients after cold exposure [34]. However, the major limitations of those studies is the small number of subjects enrolled (6 cases with BAT activation in the study by Lapa et al. and 10 patients in the study by Broeders et al), therefore the interpretation of data need to be drawn with caution. In the larger group of patients with subclinical or overt hypothyroidism (n=42) it was demonstrated that cold-induced thermogenesis was decreased in hypothyroid patients and that sufficient replacement of THs restores adaptive thermogenesis [35]. In the recent study by Maushart et al. BAT activity assessed by 18F-FDG uptake in PET-CT after cold exposure was related to levels of free T4. It was shown that cold-induced thermogenesis was approximately fourfold higher in subjects in the highest tertile of fT4 when compared to the lowest tertile of fT4 [36]. „
Minor points
Thermogenicenes: word not existing
OUR ANSWER: We corrected on „thermogenesis”
There are still a high number of typos, many slips of the pen and frequent issues with English grammar, syntax and style which need to be fixed.
OUR ANSWER: Translation into English was verified by Polish-English translator
We hope we addressed all the questions raised and the modified paper will be accepted for publication in International Journal of Environmental Research and Public Health.